# Association of Serum 25(OH)D with Metabolic Syndrome in Chinese Women of Childbearing Age

**DOI:** 10.3390/nu14112301

**Published:** 2022-05-30

**Authors:** Xiaoyun Shan, Xiayu Zhao, Siran Li, Pengkun Song, Qingqing Man, Zhen Liu, Yichun Hu, Lichen Yang

**Affiliations:** Key Laboratory of Trace Element Nutrition of National Health Committee, National Institute for Nutrition and Health, Chinese Center for Disease Control and Prevention, Beijing 100050, China; shanxiaoyun0924@163.com (X.S.); zhaoxiayuzxy@163.com (X.Z.); sirancara@163.com (S.L.); songpk@ninh.chinacdc.cn (P.S.); manqq@ninh.chinacdc.cn (Q.M.); liuzhen@ninh.chinacdc.cn (Z.L.); huyc@ninh.chinacdc.cn (Y.H.)

**Keywords:** vitamin D, metabolic syndrome, gene polymorphism

## Abstract

Objective: To analyze the associations between serum 25(OH)D levels and the risk of metabolic syndrome (MetS) and its components, and the related genetic and non-genetic factors in non-diabetic women of childbearing age in China. Methods: Subjects were randomly selected from the 2015 Chinese Adult Chronic Disease and Nutrition Surveillance. The data of sociodemographic characteristics and lifestyle factors were obtained through questionnaire survey. Anthropometry was measured by trained interviewers, and fasting blood was collected to test 25-hydroxyvitamin D [25(OH)D], total cholesterol (TC), triglycerides (TG), low-density lipoprotein cholesterol (LDL-C), high-density lipoprotein cholesterol (HDL-C), fasting blood glucose (FBG), and other related parameters. Generalized linear mode and multivariate logistic analysis were performed to analyze the associations between serum 25(OH)D and MetS and its components, adjusting for the possible confounders. Results: Body mass index (BMI), serum alanine aminotransferase (ALT), hypersensitive C-reactive protein (hs-CRP), 25(OH)D, phosphorus (P), and parathyroid hormone (PTH) levels were associated with the number of MetS’s components. G allele carriers of GC rs2282679 had higher diastolic blood pressure (DBP) and FBG levels compared with the TT genotypes, while higher genetic risk score (GRS) seemed to be associated with reduced HDL-C level. The odds ratio (OR) for MetS in lowest group of 25(OH)D was 1.533 (0.980–2.399) after adjusting for season, district, area type, latitude, age, BMI, PTH, P, ALT, CRE, interleukin-6 (IL-6), and hs-CRP, compared with the median group, but the association was not significant. An insufficient 25(OH)D concentration (<14.22 ng/mL) was significantly related to the risk of elevated waist circumference (WC) (OR = 1.612 (1.014–2.561)) and TG (OR = 2.210 (1.318–3.706)), and reduced HDL-C (OR = 1.639 (1.206–2.229)) after adjusting for the confounders among these women. Moreover, these relationships were not affected by vitamin D metabolism-related gene polymorphisms. Conclusion: After comprehensively considering various influencing factors, significant associations between insufficient serum 25(OH)D and MetS‘s components, including elevated WC, TG, and reduced HDL-C, were observed. However, MetS, hypertension, and hyperglycemia were not found independently associated with 25(OH)D levels.

## 1. Introduction

Vitamin D, a fat-soluble prohormone, exerts crucial roles in maintaining bone health and calcium homeostasis [1]. 25-hydroxyvitamin D [25(OH)D], the main form of circulating vitamin D, is widely applied as an indicator of vitamin D status. It is now well demonstrated that vitamin D also plays a role in many extra-skeletal health issues [2], such as metabolic syndrome (MetS) [3,4], type 2 diabetes mellitus (T2DM) [5], abdominal obesity [6,7], hypertension [8,9], and dyslipidemia [10].

MetS, a complex of metabolic abnormalities, is characterized by abdominal obesity, hypertension, hyperlipidemia, and elevated serum glucose [11]. Epidemiological evidence shows that, MetS is the risk factor of T2DM [12], cardiovascular disease (CVD) [13], kidney disease [14], and cancer [15]. It has been estimated by International Diabetes Federation (IDF) that one-quarter of the world’s population has MetS [16]. MetS is common in China as well, with prevalence of MetS increasing with age from 3.60% to 21.68% [17]. Therefore, MetS has attracted more and more attention in the field of public health.

Recently, many epidemiological studies are focusing on the association between vitamin D and MetS, but the results were inconsistent. For instance, a Japanese study suggested that higher 25(OH)D levels were associated with decreased risk of MetS among adults, but there were no statistically significant associations between 25(OH)D and each component of MetS after adjustment [18]. Similarly, a cross-sectional study for non-diabetic young adults in Northern Taiwan showed an inverse association of vitamin D status with MetS and its individual components, but body mass index (BMI) and homeostasis model assessment of insulin resistance (HOMA-IR) could largely remove the association [19]. Similar studies have also been carried out in the Chinese Mainland. A study for middle-aged men in Dalian in northeast China reported that decreased serum 25(OH)D level was significantly correlated with MetS [20]. In Shanghai, women without MetS were found to have significantly higher 25(OH)D level than those with MetS, but no such difference was observed for men [21]. Data based on the Chinese Longitudinal Healthy Longevity Survey (CLHLS) showed that higher serum vitamin D concentrations were associated with a lower prevalence of MetS, but the association was only statistically significant among old people with noncentral obesity [22]. Taken together, there is a lack of conclusive evidence for the Chinese population, especially for young women, to demonstrate the association between serum 25(OH)D levels and MetS [23]. Additionally, most of these studies did not consider the comprehensive effects of genetic factors, hepatic and renal function, inflammatory status, and other factors.

Thus, in the present study, we collected data from the 2015 Chinese Chronic Diseases and Nutrition Survey (CCDNS) and measured the relevant clinical parameters. We aimed to evaluate the independent association between serum 25(OH)D and the risk of MetS and its components, and analyze the related genetic and non-genetic factors, among Chinese childbearing aged women who were not diagnosed with diabetes.

## 2. Materials and Methods

### 2.1. Subjects

In this study, women of childbearing age were randomly selected from the 2015 CCDNS sample bank, according to the sampling method described in our previous reports [24]. We excluded the samples with diabetes, missing information, poor blood quality (hemolysis), and biological index values below or above the detection limit. Thus, a total of 1505 women were finally included in this study. Written informed consent was obtained from every participant. The study was conducted in accordance with the Declaration of Helsinki, and the protocol was approved by the Ethics Committee of the National Institute of Nutrition and Health, Chinese Center for Disease Control and Prevention (file number 201519-A).

### 2.2. Data Collection and Variable Classifications

The data of sociodemographic characteristics (age, nationality, and education) and lifestyle factors (cigarette smoking and alcohol consuming) were obtained by professional interviewers through questionnaire survey. The education level was divided into primary (primary school and below), medium (junior middle school/Senior High School), and advanced (junior college and above). Cigarette smoking status was divided into never and current/former smoker. If participants had a history of alcohol intake in the past year, they were classified as alcohol consumers. Height, weight, waist circumference (WC), and blood pressure (BP) including systolic blood pressure (SBP) and diastolic blood pressure (DBP) were measured by trained interviewers. BP was measured three times, and the average was used in the analysis. BMI was calculated as weight (kg) divided by height squared (m^2^). BMI levels were categorized as underweight (<18.5 kg/m^2^), normal weight (18.5–23.9 kg/m^2^), overweight (24.0–27.9 kg/m^2^), or obese (≥28.0 kg/m^2^) based on Chinese criteria [25].

### 2.3. Laboratory Measurements

Fasting venous blood (10 mL) was drawn from each participant and centrifuged and separated into plasma, serum, and red blood cells, and then stored at −80 °C in freezers until analysis. All blood samples were analyzed in strict accordance with the manufacturer’s protocol and standard laboratory test methods. Serum 25(OH)D was measured by the liquid chromatography-tandem mass spectroscopy (LC-MS/MS) (AB Sciex Pte. Ltd., Framingham, MA, USA). Parathyroid hormone (PTH) and interleukin-6 (IL-6) were measured by electronic chemiluminescence immunoassay (Roche e601, F Hoffmann-La Roche Ltd., CH4002, Basel, Switzerland). Serum fasting blood glucose (FBG), high-density lipoprotein cholesterol (HDL-C), low-density lipoprotein cholesterol (LDL-C), total cholesterol (TC), and triglyceride (TG) were measured using an automatic biochemical analyzer (Hitachi 7600, Tokyo, Japan). Serum calcium (Ca) was detected by inductively coupled plasma mass spectrometry (ICP-MS, PerkinElmer, NexION 350, Waltham, MA, USA). In addition, serum phosphorus (P), creatinine (CRE), alanine aminotransferase (ALT), hypersensitive C-reactive protein (hs-CRP) was detected by using a Hitachi 7600-210 ISE automatic biochemistry analyzer (Hitachi 7600, Japan).

### 2.4. Definitions of MetS

According to the 2017 version of the Guidelines for the Prevention and Treatment of Type 2 Diabetes in China (the 2017 version) and a joint multinational interim statement [26], MetS was diagnosed when there were three or more of the following components: WC > 85 cm; serum TG concentration > 1.7 mmol/L or with treatment for the lipid abnormality; HDL-C concentration < 1.3 mmol/L or with treatment; BP ≥ 130/85 mmHg or with treatment; or serum FBG concentration ≥ 5.6 mmol/L or previously diagnosed with T2DM.

### 2.5. SNP Selection and Detection

Based on the findings from previous studies [27,28,29], vitamin D metabolism relatedsingle nucleotide polymorphisms (SNPs), such as cytochrome P450-2R1 (CYP2R1) rs12794714, VDBP (GC) rs2282679, and vitamin D receptor (VDR) rs2228570 were selected. The SNPs selected were all in accordance with Hardy– Weinberg equilibrium (secondary allele frequency (MAF) > 0.1, Appendix A). The SNPs were detected by the improved multiple ligase detection reaction (iMLDR). A genetic risk score (GRS) for 25(OH)D was the sum of the number of A alleles of rs12794714, T alleles of rs2282679, and G alleles of rs2228570 (range from 0 to 6).

### 2.6. Statistical Analysis

All data management and statistical analyses were performed with IBM SPSS Statistics 23. Non-normally distributed continuous variables were described as medians and interquartile ranges (P25 and P75). Non-parametric statistics were used when comparing continuous variables between groups. Categorical variables were described as percentages and compared by using the chi-squared test. Associations between serum 25(OH)D and risk factors of MetS components, Ca, P, PTH, ALT, CRE, and other factors were analyzed by using the generalized linear mode. Participants were categorized into 3 levels according to our previous study [24], and the median level (14.22–18.06 ng/mL) was taken as the reference. Multivariate logistic analysis was applied to analyze the relationship between serum 25(OH)D status and risk of MetS as well as its components. The crude model was not adjusted. The adjusted model was adjusted for season, district, area type, latitude, age, BMI, PTH, P, ALT, CRE, IL-6, hs-CRP, with or without GRS. *p* < 0.05 was considered statistically significant.

## 3. Results

### 3.1. Characteristics of the Studied Population

A total of 1505 participants were included in this study, and their sociodemographic characteristics, lifestyle factors and SNPs were shown in Table 1.

### 3.2. Clinical Characteristics According to the Number of MetS’s Components

As the number of MetS components increased, there was a significantly positive association observed in BMI, WC, BP, PTH, ALT, hs-CRP, TG, LDL-C, and FBG (*p* < 0.05 for trend). On the contrary, a significant inverse association was noted between P and HDL-C and the MetS components. When the one or more components appeared, 25(OH)D decreased significantly. However, with the increase of MetS components, the trends of serum Ca, CRE, and IL-6 were not statistically significant (Table 2).

### 3.3. Relationships of Vitamin D Related SNPs and the MetS’s Components

As shown in Table 3, subjects with genotypes GG and GT of rs2282679 polymorphism had significantly lower levels of 25(OH)D and higher levels of DBP and FBG, compared to genotype TT carriers. Additionally, the GG carriers of rs2228570 (FokI) had lower 25(OH)D levels among this population. Moreover, significantly lower 25(OH)D and HDL-C level in subjects with GRS between 4–6 was observed. However, there were no differences in these parameters between rs12794714 genotypes.

### 3.4. Linear Relationships between Serum 25(OH)D and Risk Factors of MetS and Other Related Factors

The generalized linear model showed the partial correlation coefficients between serum 25(OH)D and the indicators of MetS and the other related factors (Table 4). Overall, 25(OH)D was positively associated with BMI and FBG. In addition, season, district, area type, GRS, latitude, age, PTH, CRE, IL-6, and hs-CRP were also independent influencing factors of vitamin D levels. However, no statistically significant correlations between 25(OH)D and other factors, including Ca, P, ALT, WC, TC, TG, LDL-C, HDL-C, and BP, were found.

### 3.5. Associations between BMI, MetS’s Components and Vitamin D Insufficiency

Just as shown in Table 5, vitamin D insufficiency was defined as 25(OH)D < 14.22 ng/mL according to our previous results [24], and a total of 358 (23.79%) women recruited had MetS. Participants with elevated TG and reduced HDL-C were found to have higher prevalence of vitamin D insufficiency (*p* < 0.05), but participants with hyperglycemia were more prone to vitamin D sufficiency (*p* < 0.05). The other components and BMI levels were not shown to be associated with vitamin D insufficiency (*p* > 0.05).

### 3.6. Odds Ratios for MetS and Its Components in Different Levels of Vitamin D

The results of the multivariate logistic analysis of the associations between serum 25(OH)D status and the incidents of MetS and its components are summarized in Table 6. The adjusted models were adjusted for season, district, area type, latitude, age, BMI, PTH, P, ALT, CRE, IL-6, hs-CRP, or GRS, which had been shown to be related to vitamin D or MetS. In the crude model, there were no associations between the vitamin D status and incidents of MetS, elevated waist, hypertension, and hyperglycemia. However, a U-shape trend was observed between serum vitamin D levels and elevated TG (*p*_1_ = 0.002, *p*_2_ = 0.042). After adjustments, the ORs and 95% CI for elevated WC, elevated TG, and reduced HDL-C comparing the lowest level (<14.22 ng/mL) versus the middle level (14.22–18.06 ng/mL) of serum 25(OH)D was 1.612 (1.014–2.561), 2.210 (1.318–3.706), 1.639 (1.206–2.229), respectively. No significant associations were observed between 25(OH)D levels and the incidence of MetS, hypertension, and hyperglycemia, even after adjusting for the confounding factors. Adjusting for GRS or not had no effects on these associations.

## 4. Discussion

Vitamin D deficiency is very common in female adults between 18–44 years old in China [30], and the prevalence of MetS in Chinese women aged ≥15 years old is high, reaching to 27.0% [31]. Thus, in this study, we analyzed the association between serum 25(OH)D and MetS and its components and explored the related factors for Chinese women of childbearing age (18–44 years old) based on the 2015 CCDNS.

In evaluating the association between vitamin D and MetS, we primarily analyzed the genetic and non-genetic factors that may be related to vitamin D or MetS. Though the exact underlying etiology of MetS and its components is still not well demonstrated, insulin resistance, obesity, adipose tissue dysfunction, proinflammatory states, and genetic factors are generally considered to be the contributing factors [32,33]. In our study, we also found that the BMI and hs-CRP increased significantly with the accumulation of MetS’s components. As vitamin D can be sequestered in adipose tissue, especially in obese individuals, and vitamin D deficiency can inhibit β cells from converting pro-insulin into insulin [34,35], the observed associations between vitamin D as well as its related factors, such as serum P, and MetS’s components in our study can be well understood. Moreover, experimental evidence suggests that adipose tissue, the arterial vascular wall, cardiac muscle cells, and adrenal cortex cells may be the non-classical target organs of PTH. This is why PTH may play roles in regulating body energy, blood pressure, and metabolism [36]. As found in our study, PTH was significantly higher in the group containing ≥1 components of MetS. In addition, studies estimate that a certain proportion of some individual MetS components and the collective MetS phenotype is attributed to genetic heritability [37], and some studies have reported the relationship between vitamin D metabolism-related polymorphisms and MetS. For example, Oh, JY, and Barrett-Connor, E found that the bb genotype of BsmI VDR polymorphism was associated with insulin resistance after adjustment for age, sex, BMI, and Ca and vitamin D use in a Caucasian population without diabetes [38]. Another study from Wroclaw also found that the BsmI VDR polymorphism was able to influence BMI and WC, while the FokI (rs2228570) VDR polymorphism appeared to affect insulin sensitivity and serum HDL-C level [39]. Moreover, individuals without MetS and with heterozygosis for FokI C > T presented higher concentration of TG and lower HDL-C than those without this polymorphism [40]. In addition, GC polymorphisms, such as rs2282679 and rs4588, were also shown to be associated with MetS susceptibility among the Chinese rural population, which might be through regulating the blood lipid (TG and HDL-C) levels [41]. In our study, rs2282679 was found to affect DBP and FBG. Then, we comprehensively considered the effects of three key gene polymorphisms, CYP2R1 rs12794714, GC rs2282679, and VDR rs2228570, which have been demonstrated to be related with vitamin D levels or distributions, by calculating GRS. GRS was found to be related to lower HDL-C. Moreover, in the linear regression analysis, season, district, area type, GRS, latitude, age, BMI, PTH, P, CRE, IL-6. and hs-CRP were independently associated with serum 25(OH)D. Thus, we intended to include these factors as confounders in assessing the associations between vitamin D and the risk of MetS and its components.

At present, a large number of epidemiological studies have explored the associations between vitamin D and MetS for different populations. Primarily, cross-sectional studies from different countries have shown the inverse relationship between vitamin D levels and the risk of developing MetS [20,42,43]. However, in Berlin, the independent inverse associations between vitamin D insufficiency and MetS among older adults without diabetes, could be removed by BMI [44]. Similarly, among non-diabetic young adults in Taiwan, the OR of having MetS was negatively associated with vitamin D levels after adjusting for age and sex. However, obesity and insulin resistance could affect the association [19]. In addition, longitudinal studies have also been published on the relationship between serum vitamin D levels and MetS risk, which also showed inconsistent conclusions [45,46,47,48]. However, a systematic review and meta-analysis of prospective studies concluded that vitamin D status at baseline in apparently healthy adults was inversely associated with future risks of T2DM and MetS [49]. In addition, a dose-response meta-analysis of cross-sectional studies and cohort studies among adults demonstrated that a 25 nmol/L (or 10 ng/mL) increment in the serum 25(OH)D concentration was associated with 20% and 15% lower risks of MetS, respectively [4]. Additionally, in postmenopausal women with VD deficiency, supplementation with 1000 international unit (IU) vitamin D3 alone for 9 months was associated with a reduction in the MetS risk profile [50]. In this cross-sectional study, to better assess the relationship of vitamin D and MetS, we firstly compared the prevalence of vitamin D insufficiency in subjects with or without MetS or its components. Women with elevated TG and reduced HDL-C turned out to have a higher prevalence of vitamin D insufficiency, which could be well-supported by the current theories. Then, we comprehensively considered the meaningful genetic and non-genetic factors related to vitamin D and MetS. We also found that vitamin D insufficiency resulted in about a 1.5-fold increase in the risk of MetS. However, the association was not statistically significant, which might be related to the interactions between different factors.

With regard to the relationships between vitamin D and the components of MetS, the findings from different studies were inconsistent, which may be related to race, age, and the confounders considered in the data analyses. The Furukawa Nutrition and Health Study among Japanese adults suggested that there was inverse but non-significant associations between 25(OH)D and each component of MetS [18]. The CLHLS for the elderly in 2017 indicated that adequate vitamin D levels were protective factors for elevated TG and reduced HDL-C after adjustments for other components. However, no significant association was found between vitamin D deficiency and elevated WC [22]. A cross-sectional study, conducted in Wuhan City, China, also observed significant inverse relationships between 25(OH)D and elevated TG, reduced HDL-C, and elevated BP, but not others [51]. Interestingly, for the American participants, a significant inverse relationship was observed between HDL-C and 1,25(OH)_2_D, but not 25(OH)D, suggesting that the active form of vitamin D, 1,25(OH)_2_D, might play a role in metabolic regulation [42]. A follow-up study conducted among healthy adolescent girls in Iran reported that there were significant reductions in DBP, WC, and serum FBG, TC, and LDL-C after the 9-week vitamin D supplementation, but no significant effects were observed on BMI, SBP, serum HDL-C, or TG [52]. Moreover, a recent meta-analysis including nine cross-sectional studies indicated that abdominal obesity, high BP, high TG, and reduced HDL-C were statistically associated with vitamin D status in the female population [53]. In our study, elevated WC and TG and reduced HDL-C were associated with insufficient vitamin D among women of childbearing age. No associations between vitamin D and hypertension and hyperglycemia were observed. Similar to MetS, the associations between MetS’s components and vitamin D were not found to be influenced by GRS. Of course, to verify these conclusions, case-control, cohort studies, and randomized controlled trials are needed in the future.

As this study was based on a nationally cross-sectional survey, it could well represent the current status of women of childbearing age in China. Additionally, questionnaire investigation, body measurements, blood samples collection, and laboratory measurements in the study were conducted in strict accordance with the procedures. The associations between vitamin D and MetS and its components were analyzed after adjusting for possible confounders, including season, district, area type, latitude, age, BMI, PTH, P, ALT, CRE, IL-6, hs-CRP, or GRS. Meanwhile, some limitations should also be noted. We did not include the dietary vitamin D and Ca intake to evaluate the association between 25(OH)D and incidence of the MetS. In addition, as this is a cross-sectional study, the causal relationship between vitamin D and metabolic syndrome and its components cannot be well-verified. Moreover, some other confounding information such as exercise and history of disease was not collected. Thus, further studies should be conducted to infer causality effectively, and investigate the pathophysiologic mechanisms of vitamin D for predicting MetS and other metabolic diseases.

## 5. Conclusions

The present cross-sectional study suggested that serum 25(OH)D, P, PTH, and some other common risk factors levels were associated with the number of MetS components in women of childbearing age. DBP and FBG were found to be related to GC rs2282679, while HDL-C was related to GRS. Serum 25(OH)D level was linearly correlated with BMI and FBG. In addition, we used the threshold established in our previous study to analyze the association of vitamin D and MetS. After adjustment for the possible confounders in the multivariate logistic analysis, lower 25(OH)D level (<14.22 ng/mL) was found to increase the risk of MetS’s components, including elevated WC, TG, and reduced HDL-C. However, no significant association was found between 25(OH)D levels and risk of MetS, hypertension, and hyperglycemia. Furthermore, GRS had no impact on the results of MetS risk assessment.

## Figures and Tables

**Table 1 nutrients-14-02301-t001:** Characteristics of the studied population (medians and interquartile ranges (P25–P75)).

Parameters	Median (P25–P75) or *N* (%)
**Sociodemographic characteristics**	
Age (years)	30.04 (23.98–37.81)
Nationality	
Han	1309 (86.98)
Ethnic minorities	196 (13.02)
Education	
Primary	375 (24.92)
Medium	876 (58.21)
Advanced	254 (16.88)
Area type	
Rural	891 (59.20)
Urban	614 (40.80)
Latitude (°N)	32.26 (27.78–38.02)
Distric	
Eastern	522 (34.68)
Central	478 (31.76)
Western	505 (33.55)
Season	
Spring	118 (7.84)
Autumn	774 (51.43)
Winter	613 (40.73)
**Lifestyle**	
Cigarette current/former smoker	22 (1.46)
Alcohol consumer	322 (21.40)
**SNP**	
CYP2R1 rs12794714	
AA	219 (14.55)
GA	711 (47.24)
GG	575 (38.21)
GC rs2282679	
GG	152 (10.10)
GT	620 (41.20)
TT	733 (48.70)
VDR rs2228570	
AA	333 (22.13)
GA	748 (49.70)
GG	424 (28.17)

**Table 2 nutrients-14-02301-t002:** Clinical characteristics according to the number of MetS’s components.

Parameters	Total (*N* = 1505)		MetS Components		*p*
		0 (*N* = 537)	1–2 (*N* = 812)	≥3 (*N* = 156)	
Anthropometry					
BMI (kg/m^2^)	22.67 (20.29–25.12)	20. 83 (19.29–22.48) *	23.50 (20.87–25.84) *	26.84 (25.14–29.35) *	<0.001
WC (cm)	75.40 (69.88–82.30)	70.95 (66.50–74.38) *	78.00 (71.89–84.00) *	87.05 (83.26–91.4) *	<0.001
SBP (mmHg)	115.00 (107.67–123.33)	111.33 (105.67–118.67) *	116.00 (108.33–123.67) *	128.33 (117.67–141.50) *	<0.001
DBP (mmHg)	71.33 (65.67–76.67)	69.33 (64.33–74.50) *	71.67 (66.00–76.67) *	80.67 (72.67–89.33) *	<0.001
Biochemistry					
Serum 25(OH)D (ng/mL)	16.63 (11.96–22.55)	17.15 (12.56–23.50) ^a^	16.19 (11.80–22.18)	16.35 (11.52–22.52)	0.041
Serum PTH (pg/mL)	34.21 (25.99–44.37)	31.81 (24.40–41.56) *	35.57 (26.70–46.13)	36.39 (28.50–48.90)	<0.001
Serum Ca (mmol/L)	2.26 (2.15–2.37)	2.26 (2.16–2.37)	2.26 (2.15–2.36)	2.26 (2.14–2.40)	0.510
Serum P (mmol/L)	1.35 (1.23–1.48)	1.37 (1.25–1.49)	1.35 (1.23–1.47)	1.27 (1.15–1.46) *	<0.001
ALT (U/L)	12.41 (9.00–16.31)	11.53 (8.00–14.97) *	12.98 (9.00–16.92) *	14.64 (10.00–19.00) *	<0.001
CRE (nmol/L)	76.70 (71.00–82.00)	76.00 (70.99–82.81)	76.00 (72.00–82.00)	75.92 (70.00–81.00)	0.168
IL6 (pg/mL)	1.50 (1.50–1.73)	1.50 (1.50–1.61)	1.50 (1.50–1.77)	1.50 (1.50–1.92)	0.158
hs-CRP (mg/L)	0.62 (0.21–1.44)	0.39 (0.15–0.98) *	0.69 (0.24–1.62) *	1.14 (0.45–2.19) *	<0.001
TC (mmol/L)	4.20 (3.71–4.78)	4.30 (3.86–4.86)	4.10 (3.56–4.68) *	4.24 (3.80–4.95)	<0.001
TG (mmol/L)	0.88 (0.63–1.22)	0.71 (0.54–0.93) *	0.94 (0.69–1.26) *	1.94 (1.46–2.64) *	<0.001
LDL-C (mmol/L)	2.46 (2.03–2.96)	2.37 (2.00–2.82)	2.46 (2.00–2.97)	2.70 (2.30–3.26) *	<0.001
HDL-C (mmol/L)	1.30 (1.12–1.51)	1.52 (1.39–1.67) *	1.18 (1.05–1.34) *	1.06 (0.91–1.18) *	<0.001
FBG (mmol/L)	4.88 (4.54–5.17)	4.78 (4.48–5.06) *	4.90 (4.56–5.19) *	5.16 (4.80–5.60) *	<0.001

Abbreviations: BMI, body mass index; WC, waist circumference; SBP, systolic blood pressure; DBP, diastolic blood pressure; 25(OH)D, 25-hydroxyvitamin D; PTH, parathyroid hormone; Ca, corrected calcium by albumin; P, phosphorus; ALT, alanine aminotransferase; CRE, creatinine; IL-6, interleukin-6; hs-CRP, high-sensitivity C-reactive protein; TC, total cholesterol; TG, triglycerides; LDL-C, low-density lipoprotein cholesterol; HDL-C, high-density lipoprotein cholesterol; FBG, fasting blood glucose. *: compared with the other two groups; ^a^: comparing group “0” with group ”1–2”, *p* < 0.05.

**Table 3 nutrients-14-02301-t003:** Relationships of vitamin D related SNPs and the MetS’s components.

Genotypes		25(OH)D (ng/mL)	BMI (kg/m^2^)	WC (cm)	SBP (mmHg)	DBP (mmHg)	TG (mmol/L)	HDL-C (mmol/L)	FBG (mmol/L)
rs12794714	AA (N = 219)	15.47 (11.82~20.41)	22.30 (20.39~24.77)	75.10 (70.05~80.25)	115.00 (107.00~123.67)	70.67 (65.00~77.00)	0.89 (0.67~1.25)	1.26 (1.09~1.50)	4.87 (4.50~5.14)
	GA (N = 711)	16.68 (12.07~21.92)	22.63 (20.09~25.35)	75.05 (68.08~82.50)	114.67 (107.67~123.33)	71.67 (65.67~76.67)	0.86 (0.62~1.22)	1.30 (1.12~1.51)	4.90 (4.56~5.19)
	GG (N = 575)	16.90 (11.87~24.02)	22.89 (20.52~25.11)	76.50 (70.50~82.70)	115.67 (107.67~123.33)	71.00 (65.67~76.67)	0.90 (0.63~1.22)	1.32 (1.13~1.53)	4.86 (4.52~5.17)
	*p*	0.065	0.239	0.201	0.795	0.683	0.744	0.376	0.224
rs2282679	GG (N = 152)	14.11 (10.68~18.91)	22.95 (20.02~25.50)	75.20 (68.29~83.05)	115.17 (109.42~123.25)	73.00 (65.67~77.67)	0.92 (0.63~1.28)	1.27 (1.12~1.48)	4.90 (4.56~5.19)
	GT (N = 620)	16.14 (11.87~21.15)	22.70 (20.49~25.13)	76.00 (70.24~83.00)	115.00 (107.67~124.00)	71.67 (66.00~77.67)	0.89 (0.64~1.24)	1.30 (1.11~1.50)	4.92 (4.57~5.21)
	TT (N = 733)	17.73 (12.54~23.89)	22.49 (20.26~25.08)	75.05 (69.15~81.62)	114.67 (107.00~122.83)	70.67 (65.00~76.00)	0.87 (0.62~1.21)	1.32 (1.12~1.53)	4.86 (4.51~5.12)
	*p*	<0.001	0.579	0.109	0.4470	0.017	0.430	0.109	0.012
rs2228570	AA (N = 333)	17.14 (12.48~23.72)	22.46 (20.16~25.11)	74.10 (69.02~81.02)	113.67 (107.17~123.33)	71.67 (65.33~77.00)	0.90 (0.63~1.22)	1.32 (1.11~1.50)	4.88 (4.52~5.17)
	GA (N = 748)	16.93 (12.31~22.47)	22.84 (20.27~25.30)	75.68 (69.50~82.64)	115.00 (107.33~123.67)	71.17 (65.67~76.67)	0.87 (0.62~1.24)	1.32 (1.13~1.52)	4.88 (4.55~5.20)
	GG (N = 424)	15.38 (11.47~21.56)	22.44 (20.45~24.94)	76.15 (70.10~82.74)	116.00 (108.00~123.33)	71.00 (65.33~77.00)	0.89 (0.65~1.21)	1.28 (1.10~1.50)	4.87 (4.52~5.14)
	*p*	0.010	0.644	0.070	0.414	0.920	0.983	0.378	0.559
GRS	0–1 (N = 333)	18.28 (12.62~25.08)	22.47 (20.25~25.33)	75.00 (69.02~81.38)	114.67 (106.67~122.83)	70.67 (65.00~76.67)	0.88 (0.62~1.17)	1.36 (1.14~1.57)	4.86 (4.55~5.15)
	2–3 (N = 900)	16.72 (12.02~22.30)	22.78 (20.32~25.14)	75.42 (70.00~82.46)	115.00 (108.00~123.33)	71.67 (66.00~76.67)	0.88 (0.62~1.24)	1.30 (21.11~1.49)	4.88 (4.54~5.18)
	4–6 (N = 272)	14.68 (10.97~20.20)	22.41 (20.25~24.93)	76.28 (70.05~83.00)	115.67 (107.33~123.67)	70.83 (65.00~78.33)	0.89 (0.67~1.20)	1.26 (1.10~1.52)	4.90 (4.55~5.18)
	*p*	<0.001	0.692	0.380	0.381	0.388	0.710	0.003	0.570

Abbreviations: GRS, genetic risk score; 25(OH)D, 25-hydroxyvitamin D; BMI, body mass index; WC, waist circumference; SBP, systolic blood pressure; DBP, diastolic blood pressure; TG, triglycerides; HDL-C, high-density lipoprotein cholesterol; FBG, fasting blood glucose.

**Table 4 nutrients-14-02301-t004:** Generalized linear model of the associations between serum 25(OH)D and related factors.

Parameters	β	95%CI		*p*
		Lower	Upper	
Season				
Spring	−0.061	−1.211	1.088	0.917
Autumn	2.484	1.867	3.101	<0.001
Winter	0			
Area type				
Rural	2.056	1.462	2.650	<0.001
Urban	0			
District				
Eastern	2.381	1.639	3.123	<0.001
Central	2.142	1.376	2.908	<0.001
Western	0			
Education				
Primary	0.031	−1.780	1.843	0.973
Medium	0.310	−0.483	1.104	0.443
Advanced	0			
Nationality				
Ethnic minorities	0.591	−0.358	1.540	0.222
Han	0			
Cigarette smoker				
No	1.817	−0.633	4.268	0.146
Yes	0			
Alcohol consumer				
No	−0.088	−1.878	1.703	0.923
Yes	0			
GRS				
0–1	2.635	1.725	3.546	<0.001
2–3	1.344	0.577	2.110	0.001
4–6	0			
Latitude (°N)	−0.562	−0.605	−0.519	<0.001
Age (years)	0.068	0.028	0.108	0.001
BMI (kg/m^2^)	0.112	0.003	0.221	0.044
Serum PTH (pg/mL)	−0.046	−0.063	−0.028	<0.001
Serum Ca (mmol/L)	0.367	−1.072	1.806	0.617
Serum P (mmol/L)	0.571	−0.663	1.805	0.365
ALT (U/L)	0.007	−0.023	0.037	0.644
CRE (nmol/L)	0.084	0.055	0.114	<0.001
IL6 (pg/mL)	−0.157	−0.304	−0.011	0.035
hs-CRP (mg/L)	0.181	0.048	0.315	0.008
WC (cm)	0.003	−0.026	0.032	0.825
TC (mmol/L)	2.167	−0.216	4.550	0.075
TG (mmol/L)	−0.09	−0.634	0.453	0.744
LDL-C (mmol/L)	−1.936	−4.404	0.533	0.124
HDL-C (mmol/L)	−0.381	−2.96	2.197	0.772
SBP (mmHg)	0.003	−0.029	0.034	0.874
DBP (mmHg)	−0.012	−0.058	0.034	0.608
FBG (mmol/L)	0.559	0.124	0.995	0.012

Abbreviations: GRS, genetic risk score; BMI, body mass index; PTH, parathyroid hormone; Ca, calcium; P, phosphorus; ALT, alanine aminotransferase; CRE, creatinine; IL-6, interleukin-6; hs-CRP, high-sensitivity C-reactive protein; WC, waist circumference; TC, total cholesterol; TG, triglycerides; LDL-C, low-density lipoprotein cholesterol; HDL-C, high-density lipoprotein cholesterol; SBP, systolic blood pressure; DBP, diastolic blood pressure; FBG, fasting blood glucose.

**Table 5 nutrients-14-02301-t005:** The relationships of BMI, MetS, MetS’s components, and vitamin D insufficiency.

Parameters	*N* (%)	Prevalence of Vitamin D Insufficiency, *N* (%)	χ^2^	*p*
BMI			0.715	0.870
Underweight	136 (9.04)	48 (35.29)		
Normal weight	837 (55.61)	314 (37.51)		
Overweight	379 (25.18)	148 (39.05)		
Obese	153 (10.17)	56 (36.60)		
MetS			3.223	0.073
Yes	358 (23.79)	149 (41.62)		
No	1147 (76.21)	417 (36.36)		
Elevated Waist			2.884	0.089
Yes	492 (32.69)	200 (40.65)		
No	1013 (67.31)	366 (36.13)		
Elevated TG			5.921	0.015
Yes	186 (12.36)	85 (45.70)		
No	1319 (87.64)	481(36.47)		
Reduced HDL-C			16.160	<0.001
Yes	716 (47.57)	307 (42.88)		
No	789 (53.43)	259 (32.83)		
Hypertension			0.018	0.892
Yes	102 (6.78)	39 (38.24)		
No	1403 (93.22)	527 (37.56)		
Hyperglycemia			4.240	0.039
Yes	91 (6.05)	25 (27.47)		
No	1414 (93.95)	541 (38.26)		
All	1505 (100)	566 (37.61)		

Abbreviations: BMI, body mass index; MetS, metabolic syndrome; TG, triglycerides; HDL-C, high-density lipoprotein cholesterol.

**Table 6 nutrients-14-02301-t006:** Odds ratios for MetS and its components according to vitamin D status.

Index			**<** **14.22 ng/mL (*N* = 566)**		14.22–18.06 ng/mL (*N* = 288)		≥18.06 ng/mL (*N* = 651)	
		β	OR (95%CI)	*p_1_*		β	OR (95%CI)	*p_2_*
MetS	Crude ^#^	0.244	1.276 (0.912~1.786)	0.155	1	0.032	1.033 (0.739~1.443)	0.851
	Adjusted ^a^	0.427	1.533 (0.980~2.399)	0.062	1	0.331	1.393 (0.874~2.219)	0.163
	Adjusted ^b^	0.413	1.511 (0.964~2.368)	0.072	1	0.341	1.406 (0.882~2.242)	0.152
Elevated Waist	Crude ^#^	0.200	1.222 (0.902~1.655)	0.196	1	0.013	1.013 (0.751~1.368)	0.932
	Adjusted ^a^	0.477	1.612 (1.014~2.561)	0.043	1	0.384	1.467 (0.910~2.367)	0.116
	Adjusted ^b^	0.482	1.620 (1.018~2.578)	0.042	1	0.403	1.496 (0.926~2.416)	0.100
Elevated TG	Crude ^#^	0.759	2.137 (1.306~3.495)	0.002	1	0.513	1.670 (1.019~2.738)	0.042
	Adjusted ^a^	0.793	2.210 (1.318~3.706)	0.003	1	0.464	1.590 (0.936~2.703)	0.087
	Adjusted ^b^	0.784	2.190 (1.304~3.678)	0.003	1	0.487	1.627 (0.956~2.769)	0.073
Reduced HDL-C	Crude ^#^	0.421	1.524 (1.145~2.028)	0.004	1	−0.011	0.989 (0.748~1.308)	0.937
	Adjusted ^a^	0.494	1.639 (1.206~2.229)	0.002	1	0.026	1.026 (0.752~1.399)	0.872
	Adjusted ^b^	0.481	1.617 (1.188~2.202)	0.002	1	0.061	1.063 (0.779~1.452)	0.700
Hypertension	Crude ^#^	0.370	1.448 (0.773~2.714)	0.248	1	0.466	1.593 (0.865~2.934)	0.135
	Adjusted ^a^	0.410	1.506 (0.767~2.959)	0.234	1	0.237	1.267 (0.635~2.527)	0.502
	Adjusted ^b^	0.356	1.428 (0.723~2.819)	0.305	1	0.234	1.263 (0.632~2.524)	0.508
Hyperglycemia	Crude ^#^	−0.532	0.588 (0.323~1.069)	0.082	1	0.466	1.593 (0.865~2.934)	0.135
	Adjusted ^a^	−0.528	0.590 (0.314~1.108)	0.101	1	0.043	1.044 (0.570~1.912)	0.889
	Adjusted ^b^	−0.546	0.579 (0.308~1.091)	0.091	1	0.062	1.064 (0.580~1.953)	0.841

Abbreviations: β, beta coefficients; MetS, metabolic syndrome; TG, triglycerides; HDL-C, high-density lipoprotein cholesterol. ^#^ Crude: unadjusted. ^a^ Adjusted: season, district, area type, latitude, age, BMI, PTH, P, ALT, CRE, IL-6 and hs-CRP. ^b^ Adjusted: season, district, area type, latitude, age, BMI, PTH, P, ALT, CRE, IL-6, hs-CRP and GRS.

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
