# Peer review of "Association of Serum 25(OH)D with Metabolic Syndrome in Chinese Women of Childbearing Age"

_nutrients, 2022, doi:10.3390/nu14112301_

Round 1

Reviewer 1 Report

The article is relevant, the methodology is adequate and the results are well described and brings interesting findings that can be further explored, especially the associations with polymorphisms. The authors aimed to analyze the link between vitamin D levels and the risk of metabolic syndrome. However, the parameters evaluated in the study were broader, so I suggest that the objective and conclusion be reformulated to encompass everything that was evaluated. 

Reviewer 2 Report

Manuscript ID: nutrients-1720289

The authors reported a study to associate serum vitamin D levels, through 25(OH)D and metabolic syndrome in Chinese women of childbearing age. The cohort is representative and well-characterized (sociodemographic characteristics, lifestyle, SNP, clinical parameters, anthropometric features, biochemical measurements, etc). The conclusion of the present research is that a positive association was found between vitamin D deficiency and MetS and its components including waist circumference measurement, triglycerides, HDL-C after adjustment. However, there are important limitations associated to this study:

-The authors present incomplete correlations results since correlation coefficients are not provided in the manuscript.

-There is a positive association between deficiency and MetS, and this has been previously reported by many papers. Also, the associations between vitamin D deficiency and MetS risk factors such as BMI, total cholesterol, and fasting blood glucose have been extensively described. Therefore, results have been widely discussed. I miss an evaluation of vitamin D deficiency and MetS by subgroups, comparing discrimination parameters at different BMI groups, WC, TG, HDL-C, etc.
